# The Interest of a Systematic Toxicological Analysis Combined with Forensic Advice to Improve the Judicial Investigation and Final Judgment in Drug Facilitated Sexual Assault Cases

**DOI:** 10.3390/ph14050432

**Published:** 2021-05-04

**Authors:** Sarah M. R. Wille, Karolien Van Dijck, Antje Van Assche, Vincent Di Fazio, Maria del Mar Ramiréz-Fernandéz, Vanessa Vanvooren, Nele Samyn

**Affiliations:** 1Unit Toxicology, National Institute of Criminalistics and Criminology (NICC), 1120 Brussels, Belgium; Vincent.Difazio@just.fgov.be (V.D.F.); MariadelMar.RamirezFernandez@just.fgov.be (M.d.M.R.-F.); Nele.Samyn@just.fgov.be (N.S.); 2Unit Forensic Advice, National Institute of Criminalistics and Criminology (NICC), 1120 Brussels, Belgium; Karolien.Vandijck@just.fgov.be (K.V.D.); Antje.Vanassche@just.fgov.be (A.V.A.); 3Unit DNA Analysis, National Institute of Criminalistics and Criminology (NICC), 1120 Brussels, Belgium; Vanessa.Vanvooren@just.fgov.be

**Keywords:** drug facilitated sexual assault (DFSA), toxicological analysis, judicial investigation, sexual assault

## Abstract

The conviction rate in drug facilitated sexual assault (DFSA) cases is known to be very low. In addition, the potential impact of toxicological results on the case is often not well understood by the judicial authorities. The aims of this study were (1) to obtain more knowledge concerning the prevalence of incapacitating substances in DFSA cases, (2) to create a more efficient DFSA analysis strategy taking background information into account, and (3) to evaluate the potential impact of systematic toxicological analysis (STA) on the final judicial outcome. This small-scale epidemiological study (*n* = 79) demonstrates that ‘commonly-used’ illicit drugs, psychoactive medicines and ethanol are more prevalent in DFSA cases in contrast to the highly mediatized date rape drugs. Additionally, via case examples, the interest of performing STA—to prove incapacitation of the victim—in judicial procedures with mutual-consent discussions has been demonstrated as it led to increased convictions. However, more attention has to be paid to ensure a short sampling delay and to get more accurate information from the medical treatment of the alleged victim. This will improve the interpretation of the toxicological analysis and thus its applicability in a DFSA case. The future is multi-disciplinary and will certainly lead to an efficient and more cost-effective DFSA approach in which STA can impact the final judgment.

## 1. Introduction

Drug facilitated crimes (DFCs) are defined by the United Nations Office on Drugs and Crime (UNODC) as criminal acts carried out by means of administering a substance to a person with the intention of impairing their behavior, perceptions, or decision-making capacity or by taking advantage of an impaired person after voluntary intake of an incapacitating substance [1]. DFCs include robbery, money extortion, and maltreatment of the elderly, children, or mentally ill patients. Rape or other types of sexual assault are referred to as a subclass of DFC: drug-facilitated sexual assault (DFSA). Governmental statistics show a general underreporting of DFCs as several factors complicate the recording of the actual number of cases (underreporting due to, e.g., shame, fear of being judged or memory loss of the victim, long delay between the collection of biological evidence and the alleged assault) [2]. In the US, a study performed in 2009 suggests that 18% of all women have been raped during their lifetimes, whereas only one in every six of those cases were reported to law enforcement entities [3]. Although the true prevalence of rape and DFSA is unlikely ever to be fully recognized, several international studies have attempted to quantitate its incidence [4,5,6,7,8,9]. A review of several international studies has been published by LeBeau et al. [4]. In Belgium, in 2018, 1538 charges of rape and 1712 sexual assaults were reported [10]. 

If the judicial authorities start a forensic investigation in Belgium, a medical investigation occurs and the necessary samples are taken (samples after sexual aggression—SSA). Within SSA, samples for DNA analysis, and blood and urine samples for a systematic toxicological analysis (STA) are taken. As one of the five DNA-laboratories in Belgium, The National Institute of Criminalistics and Criminology (NICC) receives about 300–500 of these SSA each year. In 50% of the cases, DNA analysis is required by the judicial authorities, while in only 10% of the cases an STA is requested. Prosecutors know the impact of DNA analysis on casework; however, the impact of a toxicological analysis on the judgment largely remains unknown. Hence, the standard judicial approach starts with a request aimed at searching for seminal fluid in the SSA. However, in cases that solely rely on the victim’s and suspect’s verbal account of the events, often the judicial authorities will dismiss the claim directly if there is a lack of other evidence or witnesses, mainly because DNA analysis will not be able to support the victim’s or suspect’s declaration. In these cases, however, STA, which is a toxicological step-wise approach using screening as well as confirmation and quantification methods for a wide range of compounds, could provide more information for their final judgment. Moreover, the UNODC guidelines for adequate analysis of DFC state that it is of importance to evaluate the drug intake, either voluntarily or administered, on the behavioral capacities of the victim. A large number of psychoactive substances have the potential to alter the victim’s state of mind, with ethanol rendered the first choice due to its accessibility and widespread use [11]. Illicit drugs, psychoactive prescription drugs, and even over-the-counter medicines are also likely candidates, either consumed alone or in combination with alcohol. The resulting pharmacological effects may include relaxation, euphoria, and lack of inhibition on the one hand, and drowsiness, loss of motor function, unconsciousness, and amnesia on the other hand. As a result, the UNODC guidelines also describe the minimal analytical methodology requirements for the performance of an adequate STA using sensitive and specific screening and quantification techniques for blood, urine and hair analysis [1]. Only validated procedures based on hyphenated chromatographic and spectroscopic techniques such as liquid-chromatography (LC)-diode array detection (DAD), LC-mass spectrometry (MS), LC-MS-MS, gas chromatographic (GC)-MS and GC-MS/MS should be applied. 

The conviction rate for perpetrators accused of sexual assaults is known to be low [12]. This manuscript aimed to obtain more objective data and knowledge concerning the forensic evidence in DFSA cases. This knowledge can then be applied to laboratory and judicial flows to improve their efficacy and cost-effectiveness. The type and prevalence of incapacitating substances, the possible impact of sample choice and other aspects such as timeframe between sampling and alleged facts are also discussed. These STA results were then evaluated based on the type of case and whether or not the forensic advisor deemed it relevant to perform a toxicological analysis. Forensic advisors are generalists in forensic sciences who do not perform analyses themselves but advise magistrates about the forensic-technical possibilities within their cases, taking into account all the relevant contextual information [13]. The authors seek to obtain more knowledge, create an efficient DFSA analysis strategy and create awareness for the judicial authorities that recreational, over-the-counter (OTC’s) and prescription drugs, and ethanol can play a role in their cases. Finally, this paper aims to evaluate the potential impact of STA on the final judicial outcome. 

## 2. Results

### 2.1. Study Population

Background characteristics of the case and victims are described in Table 1 for the test set (2017–2018, *n* = 79). Of the victims, 92% are female, 5.1% male, and 2.5% gender-neutral. As Belgian law assigns more severe penalties depending on the victim’s age, we divided our victims using the same age categories as the Belgian law. In 70% of the cases, the victim was 18 years or older, 19% between 16 and 18 years old, 10% between 14 and 16 years old and 1.3% between 10 and 14 years old. There were no victims younger than 10 years old in our dataset. 

For the test set 1 (2017–2018), 26 cases consisted of a sexual assault with an unknown perpetrator, 23 cases were sexual assault investigations where the suspect admits to having seen/met the victim but denies (parts of) the sexual contact, 22 cases consisted of sexual assault where the sexual contact is admitted, but the suspects claim there was mutual consent, 8 cases could not be clearly defined (e.g., cases with multiple suspects with different statements); these cases are categorized under “other”. The FA control set 2 (2014–2015) consisted of 18 cases with an unknown author, 20 cases of denied (part of) sexual contact, 19 cases of discussion of mutual consent, and 7 others.

In 18% of the cases, the victim believed to have been administered a drug by the assaulter. In 4% of the cases the victim stated to have (voluntarily) used drugs around the time of the assault. In 46% of the cases, the victim admitted to having consumed alcohol. Only in 1 case, the victim stated that they combined alcohol and drugs. During the medical examination, the victim is questioned about medication usage in the week before the assault; this was the case in 34%. 

The time delay between sampling the blood and urine samples and the alleged incident ranged from 1 h and 44 min to 68 h. In 35% of the cases, the time delay was shorter than 8 h, with 48% collected between 8 and 24 h, and 17% greater than 24 h (but less than 4 days).

### 2.2. STA Results

In 73 out of the 79 cases, both urine and blood samples were collected. In 4 cases, only urine was collected, and in 2 cases, only blood was sampled. The STA was negative in 39% (*n* = 29) of the blood samples and in 30% (*n* = 23) of the urine samples. The detected drug classes and their prevalence in blood and urine are shown in Figure 1; there is also an indication with regards to the different categories of the STA request: proposed by the forensic advisor, as a result of a direct demand by the magistrate or simply because the case was included in our code 37 study. Table 2 describes the individual drugs and medication found as well as their blood concentrations. The blood and urine results of each individual case are presented in Table 1.

Ethanol was found in 19% (*n* = 14) and 27% (*n* = 21) of the blood and urine samples respectively. Ethanol was detected as the only compound in 11% (*n* = 8) of blood samples. In 7% (*n* = 5) ethanol was combined with medication. In urine, ethanol was detected alone (*n* = 6, 8%) or in combination with medication (*n* = 7, 9%), with drugs (*n* = 3, 4%), or drugs and medication (*n* = 5, 7%). 

One or more illegal drugs were detected in 19% of all blood samples (*n* = 14), and in 25% of urine samples (*n* = 19). Eleven % (*n* = 8) of the blood samples had one or more drugs present, while in another 7% (*n* = 5) illegal drugs were combined with medication. In 6% (*n* = 6) of urine samples, drugs were solely detected and in 10% (*n* = 10), a combination with medication was found. 

Medication (one or more classes) was found in 43% (*n* = 32) of blood samples and 48% (*n* = 37) of urine samples either alone or in combination with ethanol and or illegal drugs. For blood and urine samples respectively 29% (*n* = 21) and 18% (*n* = 17) contained solely medication. 

In 1% (*n* = 1) of blood samples and 4% (*n* = 4) of urine samples ethanol, drugs and medication were combined. 

In 26% (*n* = 19) of the cases a difference was observed between the detection in blood and urine. Most compounds were more easily detected in urine: alcohol *n* = 3, drugs *n*= 6, medication *n* = 5. In 2 cases medication was detected in blood, which was not detected in urine (Figure 2).

### 2.3. FA Interpretation 

In the FA test set (set 1), STA was proposed to the magistrate by the forensic advisor in 28 cases (36% of the total number of cases in our study) and was finally followed by the magistrate in less than half of them (only 13 cases (15%)). Thirty-seven % (*n* = 10) of the cases in which an STA was proposed was categorized as “dispute of mutual consent”. In approximately one in five (22%) of the cases where no STA was proposed, a relevant toxicological finding was obtained. When evaluating the negative STA screenings, for those cases proposed by the forensic advisor, but finally not selected by the magistrates, only 10% (*n* = 3) revealed to be negative. In the ones selected by the magistrates, 21% was negative (*n* = 6), while the other study samples were negative in 68% of the cases (*n* = 20). 

The ethanol-positive samples represented respectively 35% (*n* = 6), 6% (*n* = 1) and 53% (*n* = 9) of cases selected by FA, magistrates or other samples of the study. The drug-positive samples consisted of respectively 16% (*n* = 3), 16% (*n* = 3) and 58% (*n* = 11) of cases selected by FA, magistrates or other samples of the study. The medication-positive samples were respectively 19% (*n* = 7), 14% (*n* = 5) and 64% (*n* = 23) for cases selected by FA, magistrates or other samples in comparison to the total amount of medication positive samples. The FA combined with STA resulted in a decrease of dismissed cases from 84% in 2014–2015 (control set 2) compared to 65% in 2017–2018 (test set 1) and an increase in convictions from 11 in 2014 to 24% in 2017 (Figure 3). Table 1 shows the cases in which the FA was requested and the context information. 

## 3. Discussion

The potential impact of toxicological results on the case is often not well understood by the judicial authorities, as indicated by the difference between the number of STA recommended by the forensic advisors and the officially requested number by the magistrates (Figure 4). Therefore, the study aimed to evaluate if a contextual approach, where the victim’s statement and, if known, the suspect’s statement, is taken into account via the forensic evidence, could lead to a more specific treatment of DFSA cases and finally could elevate the conviction rate.

Several factors were taken into account to evaluate whether or not to propose a toxicological analysis to the magistrate. The first important factor was the time frame in which the evidentiary biological samples were taken. Current recommendations by the Society of Forensic Toxicologists (SOFT) DFSA committee are that urine specimens should be collected within 120 h and blood within 24 h of the incident [14]. If the time between the sexual assault and the sampling exceeded these recommendations, STA of the urine and blood samples was no longer proposed. The forensic DFSA investigation can be complicated due to a delay in reporting by the victim either because of memory impairment induced by incapacitating substances and/or the traumatic experience, and the psychological issues attributed to the incident. For a toxicological laboratory, detection of the incapacitating drug can be difficult depending on the time delay between the alleged facts and the sampling, and the significant variations in pharmacokinetics of the drugs involved; toxicological findings can be biased due to the quick elimination of compounds such as GHB or ethanol. Moreover, long delays can complicate interpretation due to possible intake or administration of substances after the alleged facts, but before sampling, by the victim or administration during first aid medical treatment (e.g., case 13,23,26,43,52,58 Table 1 and Figure 2). Therefore, it is necessary to stress the importance of a correct medical file attached to the obtained samples (e.g., are substances administered for medical treatment, are compounds used voluntarily after the alleged facts?). The variable time window after drug ingestion/consumption and the sampling has to be considered when interpreting a toxicological finding; a negative result does not conclusively prove that no drugs were consumed at the time of the alleged incident, and a positive finding should be evaluated considering all of the information provided in the medical file of the victim. It is important to discuss the difficulties of interpreting a toxicological finding with the prosecutor or investigating judge. In addition, the judge should be aware of the information needed for an adequate toxicological interpretation of the STA results. Because of the time delay in the sampling and the possible low amount of consumed drugs, toxicologists should adapt laboratory methodologies to detect a wide range of compounds with adequate sensitivity. The combined use of techniques that result in a broad screening (such high-resolution mass spectrometric techniques (HRMS)), and sensitive multi-compound quantitative target methods, will result in an up-to-date screening of all possible compounds (including new psychoactive substances), while insuring the necessary sensitivity specifically needed for DFSA cases. This methodology was described and utilized in our study. It is clear, that in DFSA cases, lower concentrations found in blood and/or urine samples can still be relevant for the case interpretation in contrast to other types of case work such as, e.g., post-mortem. The UNODC guideline provides detection limits for compounds such as ethanol, GHB and analogues, benzodiazepines, barbiturates, antidepressants, OTC’s, opiates, non-narcotic analgesics, and illegal drugs [1]. It is generally acknowledged in the scientific community that urine is the most useful specimen in typical DFSA investigations as drugs and their metabolites become more concentrated in urine samples and have a longer “detection window.” This enables drugs to be more easily detectable if sampling occurs a while after the alleged incident. This is confirmed in Figure 2, illustrating that more compounds are detected in urine than in blood samples. Consequently, urine sampling is essential when there is a time delay between sampling and a potential drug exposure one to four days prior [14]. However, blood concentrations provide more information concerning the victim’s incapacity, which the prosecutor or judge often requires. This is certainly the case for the most prevalent compound in the cases, ethanol. Ethanol is rapidly eliminated from the blood; however, if detected, retrograde extrapolations can result in a better interpretation of the effects at the moment of the alleged facts. To improve the value of a toxicological investigation for the final judicial case, it is thus of utmost importance for everybody dealing with DFSA cases (medical staff and police in contact with the victim, forensic trained medical doctor, etc.) to minimize the time delay between sampling of biological samples and alleged assault.

This also links to a second factor when considering proposing an STA investigation: the victim/suspect/witness’s statements. If samples were taken within the above-mentioned time frame of 24–120 h (blood/urine) and the victim declares to have been under the influence of a drug (voluntary and/or involuntary, drugs/medicine and/or alcohol), STA was proposed. It was clear that STA was of major importance in the DFSA categories in which (1) the suspect admits to having seen/met the victim but denies (parts of) the sexual contact and (2) for sexual assaults where the sexual contact is admitted but the suspects claim there was mutual consent. Typically, when consent is disputed, DNA results will not be case informative; however, STA may be important. Certainly, in cases with declarations concerning the intake of drugs or alcohol, an STA could help determine if the victim was still capable of giving consent. However, it is not always easy to determine the STA importance in an investigation. It is clear from the study data, that not all declarations are complete (Table 1). In addition, in sexual assaults involving an unknown suspect, the forensic advisor’s toxicological analysis was not initially suggested. However, magistrates determined that if the victim states that an exogenous substance can be involved, STA should be suggested to give the victim the sense that (s)he is being heard. STA will of course never help to identify a suspect, but the obtained information could be psychological support to the victim. Of course, in cases where the suspect denies to having met/seen the victim, DNA will be the main interest in the first instance, and depending on the DNA outcome, STA can be performed in a second phase. An issue complicating the interpretation of toxicological findings, is the lack of background information. Important information that has to be collected includes estimated time between facts and specimen collection, types and quantities of voluntarily consumed alcohol and/or drugs (recreational, prescription or OTC), experienced symptoms, age of the victim, and some additional case information to allow more comprehensive toxicological conclusion [15]. The relevant case information is gathered and centralized within the FA, but it is apparent from reviewing the STA results in our study that victims do not often reveal their (in)voluntary drug or alcohol intake (Table 1). This shows that even by using a more contextual approach focusing on the suspect’s declaration, there are some cases in which toxicological analysis was not advised because there was no indication of drug/alcohol use, but where toxicological analysis could have impacted the outcome. In a study evaluating DFSA cases processed over 3 years, Scott-Ham and Burton observed that one-quarter to one-third of the alleged victims admitted to using an illicit drug [9]. In a study funded by the US National Institute of Justice the researchers estimated that less than 5% of DFSA cases involved a drug being surreptitiously administered to the victim, whereas when voluntary drug use is considered, over one-third of cases may be facilitated by drugs [16]. As observed from our data (Table 1), alleged victims tend to underreport alcohol and drug use due to feelings of self-blame or because they believe this will impact their credibility in court. In contrast, in our cases 18% (*n* = 14) of victims claimed to have consumed a ‘spiked’ drink or to have been drugged, while this could not always be confirmed via the STA data (Table 1). In case 34 (Table 1), the victim claimed to be drugged by her ex-partner via a spiked drink. The prosecutor requested an additional hair analysis of the victim after the STA demonstrated a positive MDMA blood concentration to make sure it was a single intake, as she claimed never to use MDMA. By analyzing the hair in segments of 1 cm, the possible historic drug use of the victim per month (as the average hair growth is established to be one cm/month) could be evaluated. The hair analysis demonstrated regular use over the past months and the case was dropped by the prosecutor. A clear, cost-effective flowchart provided to the judicial authorities is important to inform them of the possibilities of toxicological analysis and give them advice on how they can efficiently build up each specific case.

Our small-scale study observes that ethanol was the most prominent compound present in blood samples (Figure 1). It is important to make enforcement officers and prosecutors aware that DFSA is not automatically associated with the date-rape drugs reported in media or in the general public’s mind. The definition of DFSA is also very important, as some still eliminate ethanol as a potential DFSA drug, while others consider it a crime whenever an individual takes advantage of a person that is incapacitated by voluntary intake in order to have non-consensual sexual relations. Failure to recognize the increased risks associated with voluntary consumption of drugs like ethanol is a public concern. DFSA casework involving ethanol is complex and poses challenges due to ethanol’s pharmacokinetics and effects. It is estimated that alcohol intoxications are present in one-third to three-quarters of all sexual assault cases and are involved in approximately one-half of all sexual assaults among college students [17]. Through the FA, the importance of ethanol as a possible DFSA drug was brought to the magistrates’ attention. Prevalence’s are, however, difficult to compare across different studies as the time interval between the alleged incidence and sampling is of importance. The detection time window of course depends on the amount of intake and the metabolization rate of the person, however, the detection time of ethanol, is relatively short in comparison with most drugs and medication. Alcohol produces a wide range of effects such as confusion, dizziness, memory loss, impaired judgment, behavioral changes, cognitive impairment, reduces inhibitions, drowsiness, nausea, vomiting, loss of consciousness, coma, and death which are dose-dependent [18]. As some of ethanol’s effects include amnesia and black-outs, it may contribute to a poor recall of events and thus an extra challenge for forensic investigation and possible prosecution. In 62% of the cases in which the victim/suspect/witness’s statements referred to the victim as being intoxicated by ethanol, the time frame between the alleged facts and the blood sample collection surpassed 12 h, which may result in negative ethanol results due to elimination. As no ethanol quantitation can be provided, it will not be possible to perform retrograde extrapolation equations to calculate the ethanol concentration at the time of the incident. However, this could potentially be crucial information to the case. For a toxicologist measuring other ethanol biomarkers such as phosphatidyl-ethanol (PEth) or ethyl glucuronide (EtG), this may help to resolve detectability of ethanol use; however, for the judicial authorities, this information is not adequate to prove incapacitation [19]. In one particular judgment, 2.21 g/L (back-calculated) blood-ethanol lead to a modification of the trial outcome as the judge decided that this would result in an incapacity of the victim to give consent (case 36, Table 1). It is important to take all available information, such as the age of the victim, into consideration and clearly describe the assumptions used to extrapolate the blood-alcohol concentrations: average elimination rate, time of alleged facts versus sampling time, possible tolerance of the victim due to chronic ethanol abuse. In 19% of all blood samples, one or more illegal drug was detected (Figure 1). Even though the media have actively portrayed flunitrazepam and GHB as important ‘date rape’ drugs, these were only observed in urine (one case of each drug). The most prevalent compounds were ‘classical’ recreational drugs popular in the Belgian drug-using population: cocaine, amphetamine/MDMA and cannabis [20]. Prescription drugs were found in about 43% of the blood samples, with antidepressants, benzodiazepines, and neuroleptics being the major groups detected. Other compounds observed were opioids, painkillers (NSAID, paracetamol), hart-medication, anti-histaminic, antibacterial or antifungal medication, methylphenidate, diabetic medication. Some of these drugs have central nervous system (CNS)-depressant activities and are capable of causing sedation or incapacity, while others render a prospective victim susceptible to an assault. For reviews concerning the effects and pharmacokinetics of several prescription drugs, we refer to the reviews of Couper et al. [18], Stockham and Rohrig [21] and Montgomery [22]. Prescription drugs often are consumed as part of the victim’s treatment, but their effects or side effects can be potentialized via co-use with ethanol or illegal drugs. In our study, half of the samples containing ethanol also were found positive for an illegal or prescription drug. In some cases, lack of therapeutic compliance can also be of importance. In one case in our study (case 66, Table 1), the victim was positive for several neuroleptics and an antidepressant in urine. The toxicologist reported two possible conclusions (based on the information the laboratory had received): either the drugs were administered and therefore the effects would have incapacitated the victim or the victim was prescribed this medication for a longer period, possibly developed a tolerance and thus the concentration would normally not have led to severe incapacitation at the time of the alleged facts (sub-therapeutic to therapeutic blood concentrations). When the prosecutor investigated the case further, this victim seemed to have filed several similar complaints all-over the country. Again, this shows that getting a lot of background information is of utmost importance for a good toxicological interpretation.

While STA can be seen in most toxicological laboratories as a well-defined and routine protocol, it is clear from the significant increase in convictions and decrease of dismissed cases (Figure 3), that advice to the judicial authorities concerning the possibilities and limitations of STA, the link of background information to the STA results, and finally, aid with the interpretation of the scientific findings in a judicial system, could lead to a better knowledge of STA benefits and finally a better judicial outcome. When comparing then number of convictions between the judicial year 2018–2019, for which FA has been taken into account, versus the judicial year 2014–2015, in which FA was not yet applied, an increase of about 50% could be observed. The number of dismissed cases was decreased by about 30%. 

## 4. Materials and Methods

### 4.1. Case Selection

In Belgium, all crimes are categorized and receive a certain code, which refers to the type of crime. Cases of sexual assault such as rape, sexual exploitation, and debauchery receive code 37. From September 2017 until August 2018 (a Belgian judicial year), 1374 cases with the code 37 were registered in the Antwerp judicial district. Only the 114 cases, for which SSA were obtained, were taken into account. For 94 of them, a forensic advice (FA) was written. The remaining twenty cases with SSA were eliminated for different reasons, for example, transferred to another judicial district. The dataset of 94 cases was then further limited to the 79 cases in which the magistrate took the FA into account. For these 79 cases, an STA was performed (Figure 4). 

To evaluate the impact of the new analysis strategy on the outcome of the cases, the data set from the judicial year 2017–2018 (Set 1) was compared to the data set from the Antwerp district in 2014–2015 (Set 2). For data set 2, the cases were treated without the interpellation of a FA.

### 4.2. STA

The urine and blood SSA-samples were stored at −18 °C before analysis. In general, blood samples were collected in venutubes with sodium fluoride as the anticoagulant. STA, applied in this study, consisted of screening using high-resolution mass spectrometry (LC-HRMS) via the Xevo-G2-QTOF-XS (Waters, Manchester, UK) with UNIFI software [23] followed by several target drug methods [24,25,26,27,28,29,30,31,32] applied for confirmation and quantification [25,26,27,28,29,30,31,32,33]. A total sample volume of maximum 1 mL (100 µL for screening and 900 µL for all confirmation and quantification methods) was necessary when applying the methods as stated below. Ethanol was quantified via a headspace GC-FID, gamma-hydroxybutyric acid (GHB) via GC-MS after alkylation/acetylation with TFAA/HFB-OH [24], and the illicit drugs (amphetamines and analogues, cocaine and metabolites, opioids), as well as drugs such as antidepressants, benzodiazepines and neuroleptics were quantified via high-pressure liquid chromatographic-tandem mass spectrometric techniques (UPLC-MS/MS) in multi-reaction-monitoring mode (MRM) [25,26,27,28,29,30,31,32]. All the methods applied were validated according to international standards and published in peer reviewed journals [25,26,27,28,29,30,31,32]. The method for the quantification of neuroleptics in blood and urine was not yet published; therefore, it is described in this paragraph. The method detected the following neuroleptics and metabolites: haloperidol, olanzapine, desmethylolanzapine, levomepromazine, quetiapine, 7-hydroxy quetiapine, risperidone, 9-hydroxy-risperidone, prothipendyl, aripiprazol, clozapine, desmethyl-clozapine, amisulpride, bromperidol, sertindole, clotiapine, bemperidol, tiapride, pimozide, droperidol, pimpamperone, sulpiride and flupentixol. The sample preparation consisted of a protein precipitation of 100 µL using plasma/blood or urine and the addition of 500 µL of acetonitrile (ULC-MS grade, Biosolve, Valkenswaard, The Netherlands). After centrifugation, 1 µL of the extract was injected onto the UPLC-MS/MS (Xevo TQ-S tandem mass spectrometer, Waters, Manchester, UK). Analytes were separated using an Acquity UPLC BEH C18 (2.1 × 50 mm, 1.7 µm) (Waters). The column was kept at 55 °C, with a mobile phase flow of 0.8 mL/min and the gradient started out during the first 0.2 min with 99% of solvent A (0.1% formic acid in water) and 1% of solvent B (acetonitrile). Over the next 2.3 min, solvent B increased to 70% and then to 99% in the next 0.25 min (staying for 0.3 min). The run time with equilibration to initial conditions was 3.5 min. Ionization was achieved using electrospray in positive ionization mode (ESI+). Nitrogen was applied as nebulization and desolvation gas at a flow rate of 1000 L/h and heated to 650 °C. Capillary voltage and source block temperature were 1 kV and 150 °C, respectively. The collision gas (argon) pressure was maintained at 0.35 Pa (3.5 × 10 ^−3^ mBar) and the collision energy (eV) was adjusted to optimize the signal for the most abundant product ions, which were subsequently used for MRM analysis (Appendix A). The method was validated according to the publications of Wille et al. [33,34].

## 5. Conclusions

This small-scale epidemiological study demonstrates that ‘common’ compounds such as medication, classical drugs, and ethanol are often detected in DFSA cases compared to mediatized date rape drugs. Additionally, via case examples, the interest of STA to prove incapacitation of the victim in judicial procedures with mutual-consent discussions is proven and has resulted in an increase in condemnations. However, more attention has to be paid to ensure a short sampling delay and to get insight into the medical treatment. When obtaining more accurate information and a better sampling protocol, the interpretation of the toxicological analysis and thus its applicability will even improve. It is clear that forensic laboratories have invested a lot in very adequate and sensitive methods to detect a whole range of compounds in various biological matrices during the past decades. However, an intensive collaboration and an investment in adequately trained personnel either in the police department or a health service resulting in fast and correct sampling, a service that can ‘translate’ the forensic evidence and its importance to the judicial authorities or the ‘real’ questions of the magistrate to the forensic expert will be the way to make things move forward. The future is multi-disciplinary and will certainly lead to an efficient and more cost-effective DFSA approach in which STA can have more impact on the final judgement. 

## Figures and Tables

**Figure 1 pharmaceuticals-14-00432-f001:**
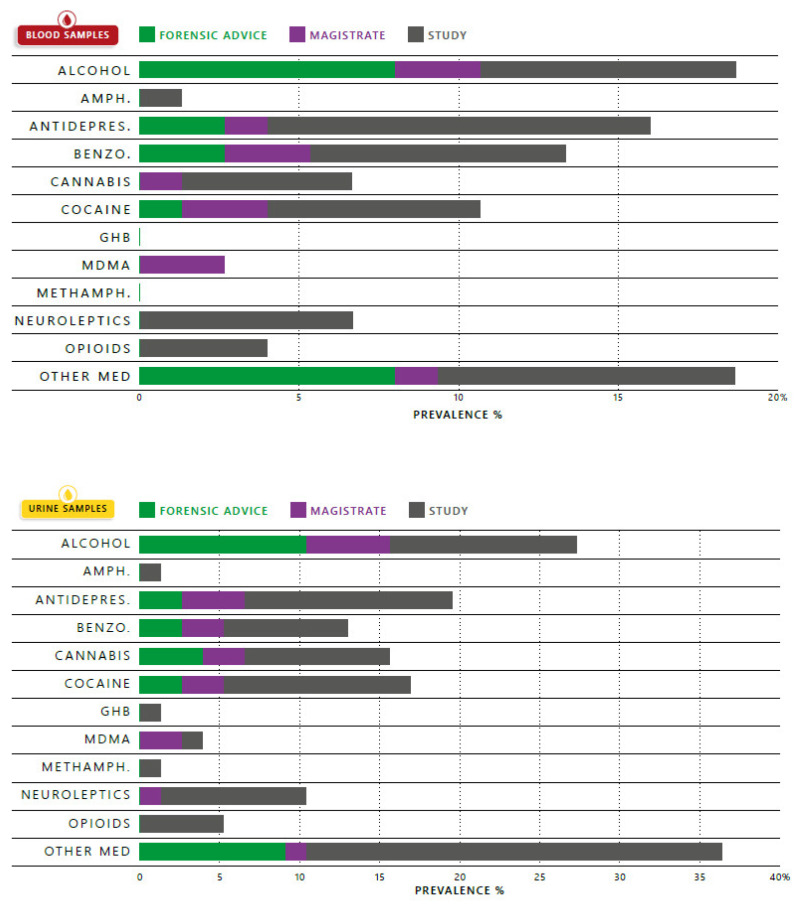
Prevalence of compounds in blood and urine samples indicated in percentage. FA: samples selected via forensic advice (*n* = 15); MAG: samples selected via magistrates (*n* = 13); study samples (*n* = 51); Amph: amphetamine; Antidepres: antidepressants; Benzo: benzodiazepines, Methamph: methamphetamine; Other Med: other medication—painkillers, heart-medication, anti-histaminics, antibacterial or antifungal medication, methylphenidate, diabetic medication.

**Figure 2 pharmaceuticals-14-00432-f002:**
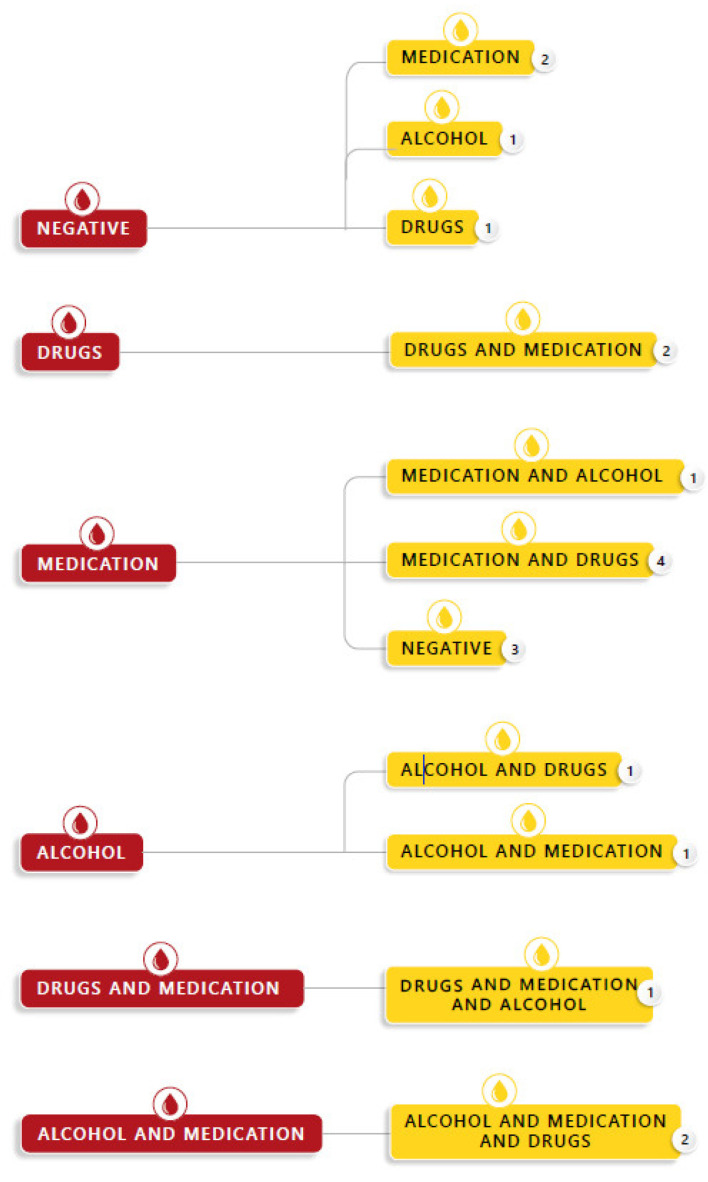
Discrepancies between compounds detected in blood and urine.

**Figure 3 pharmaceuticals-14-00432-f003:**
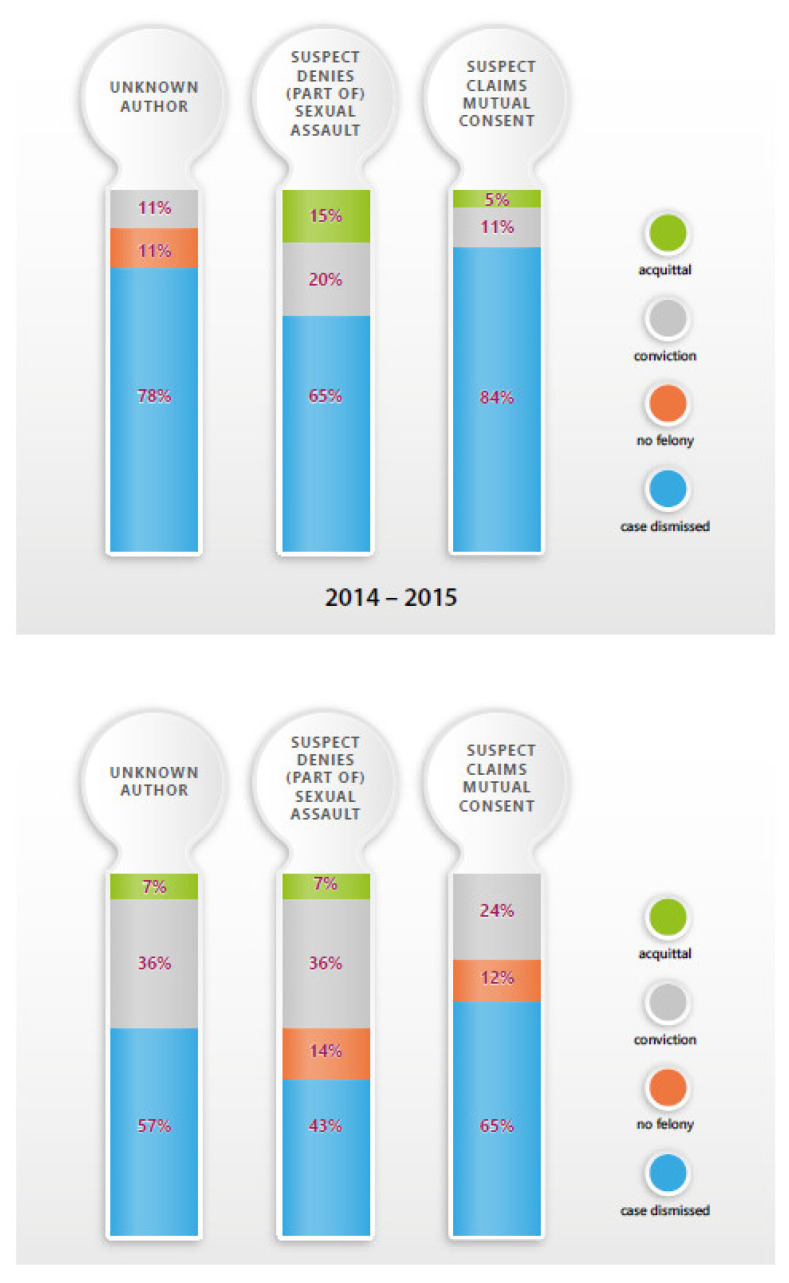
Percentages of judicial decision depending on the type of DFSA for reference year 2014–2015 without Forensic Advice and year 2018–2019 with Forensic Advice.

**Figure 4 pharmaceuticals-14-00432-f004:**
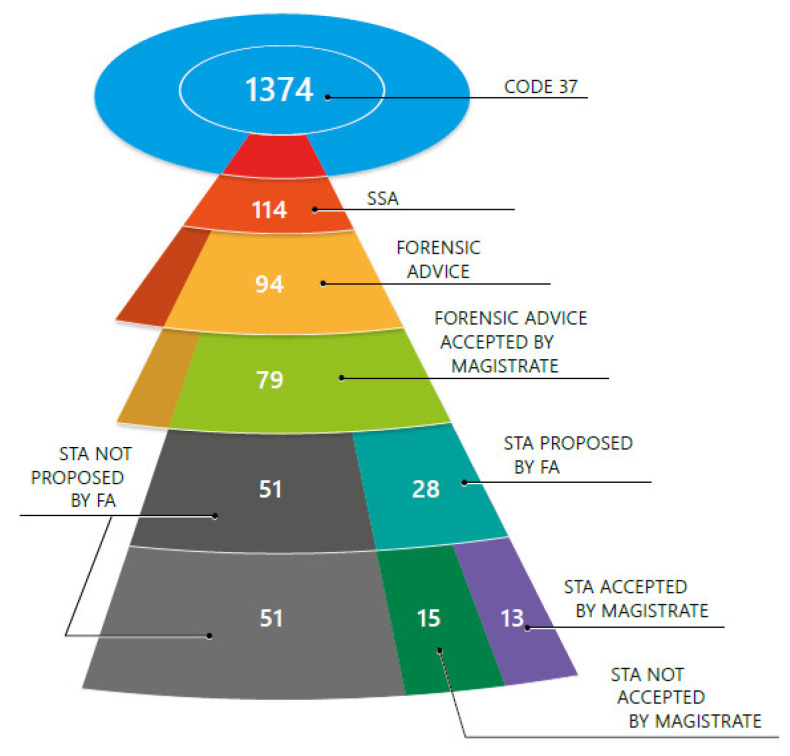
Case selection flow. The number of cases are indicated; SSA: samples after sexual aggression; STA: systematic toxicological analysis.

**Table 1 pharmaceuticals-14-00432-t001:** Background characteristics and case results.

Case N°	Age (Years)	Gender	Information According to Victim				Time between Sexual Assault and Sampling	STA Blood		STA Urine
			Suspicion of Being Drugged	Voluntary Intake				Detected	Back Calculation	
				Drugs	Medication in the Week before the Assault	Alcohol				
1	>14–16	F		■			07h40	8.9 ng/mL THC-COOH		THC-COOH
2	>14–16	F		■			08h50	5.1 ng/mL THC-COOH		Cetirizine, THC-COOH
3	>18	F			■		01h44	Negative		Negative
4	>16–18	M			■		05h10	88.2 ng/mL Fluoxetine, 264 ng/mL Norfluoxetine, 21.5 ng/mLMethylphenidate, >300 ng/mL Rilatinic acid, 34.3 ng/mL Aripiprazol		Fluoxetine, DMF, Methylphenidate, Rilatinic acid, Aripiprazol
5	>18	F			■		09h30	1.56 g/L Ethanol, Paracetamol, Bromazepam	3.09 g/L Ethanol	2.43 g/L Ethanol, Paracetamol, Bromazepam
6	>18	F	■				13h30	Negative		N/A
7	>18	F					24h00	Negative		Paracetamol
8	>18	F					44h30	Negative		Negative
9	>18	F	■			■	11h10	Negative		Aspirin, Metipranol
10	>16–18	F					06h30	Azithromycin		Azithromycin
11	>16–18	F			■		14h08	86.5 ng/mL Aripiprazol		Aripiprazol
12	>18	F					12h00	N/A		3.11 g/L Ethanol, Trazodone, m-Cpp, Cocaine, BZE, EME, CE, Levamisole, Paracetamol, Doxylamine
13	>18	M	■		■	■	68h22	138 ng/mL Nordiazepam, 3.0 ng/mL Oxazepam		Nordiazepam, Oxazepam, BZE, THC-COOH
14	>18	F	■			■	32h45	Negative		Negative
15	>18	F				■	17h00	Paracetamol		Paracetamol, Piracetam, BZE
16	>14–16	F					11h45	Negative		Negative
17	>18	F					04h20	Negative		Negative
18	>18	M					05h00	1.81 g/L Ethanol	2.56 g/L Ethanol	2.45 g/L Ethanol, THC-COOH
19	>18	F				■	14h30	Valsartan		Valsartan, Nevibolol
20	>10–14	F					30h40	Negative		Aspirin
21	>18	F					09h20	Negative		Negative
22	>18	F	■				24h00	Negative		Negative
23	>18	F	■		■	■	14h30	0.47 g/L Ethanol		0.40 g/L Ethanol, Aspirin
24	>18	F					03h00	235 ng/mL Bromazepam, 2807 ng/mL Amphetamine, 5.8 ng/mL THC-COOH		0.16 g/L Ethanol, Bromazepam, Temazepam, Amphetamine, Methamphetamine, GHB (336.2 µL/mL), THC-COOH, BZE, Paracetamol
25	>16–18	F	■			■	06h00	37.8 ng/mL Diazepam, 4.7 ng/mL THC-COOH		0.44 g/L Ethanol, Diazepam, Nordiazepam, Temazepam, THC-COOH
26	>18	F			■	■	24h00	1.09 g/L Ethanol, 19.6 ng/mL Citalopram, 5.4 DMC		0.46 g/L Ethanol, Citalopram, DMC
27	>18	F					04h00	0.38 g/L Ethanol, Naproxen, Ibuprofen	0.98 g/L Ethanol	0.72 g/L Ethanol, Naproxen, Ibuprofen, THC-COOH
28	>16–18	F					02h00	Salbutamol		Salbutamol, BZE, EME
29	>18	F					04h40	N/A		Negative
30	>18	F					03h50	24.5 ng/mL Trazodone, 9 ng/mL Dosulepin, 0.8 ng/mL Zolpidem, 6.1 ng/mL Alprazolam, 31.0 ng/mL Amisulpride		Trazodone, Dosulepin, Duloxetine, Zolpidem, Alprazolam, OH-Alprazolam, Amisulpride, Gliclazide, Bisoprolol, BZE
31	>18	F					19h10	Negatieve		Diclophenac
32	>18	F					06h20	Negative		Negative
33	>18	F			■		34h00	Negative		Negative
34	>18	F	■			■	16h00	7.5 ng/mL BZE, 166 MDMA, 17.0 MDA		0.22 g/L Ethanol, BZE, EME, Cocaine, MDMA, MDA
35	>16–18	F			■		04h20	7.9 ng/mL Mirtazapine, 0.9 ng/mL Quetiapine, 4.7 ng/mL OH-Respiridone, 4.8 ng/mL Sulpiride		Mirtazapine, Quetiapine, OH-Respiridone, Sulpride, Metformine, Paracetamol
36	>18	F			■	■	04h30	1.53 g/L Ethanol	2.21 g/L Ethanol	2.20 g/L Ethanol
37	>18	F	■			■	16h00	Negative		Negative
38	>18	F			■		16h45	N/A		Tramadol, Aspirin
39	>18	F	■			■	15h00	83.0 ng/mL MDMA, 8.0 ng/mL MDA		MDMA, MDA
40	>18	F				■	16h50	Negative		Negative
41	>14–16	F					07h00	Negative		Prometazine
42	>14–16	F					08h00	N/A		Negative
43	>18	F					50h10	0.75 g/L Ethanol		0.94 g/L Ethanol, Paracetamol, Piracetam
44	>16–18	F			■	■	11h00	1.04 g/L Ethanol	2.69 g/L	1.47 g/L Ethanol
45	>16–18	F			■	■	09h50	Negative		Negative
46	>18	F			■	■	15h50	38.4 ng/mL Sertraline, 156 ng/mL BZE, 10.2 ng/mL EME, 3.4 ng/mL CE		Sertraline, BZE, EME, CE, Cocaine, Xylometazoline
47	>18	F				■	15h00	233.4 ng/mL BZE, 8.5 ng/mL EME, 3.9 ng/mL CE		0.11 g/L Ethanol, BZE, EME, CE, Cocaine
48	>16–18	F					48h00	Negative		Negative
49	>18	F			■		02h00	Paracetamol		Paracetamol
50	>16–18	F				■	04h00	0.80 g/L Ethanol, Paracetamol, Piroxicam	1.40 g/L Ethanol	1.17 g/L Ethanol, Piroxicam, Paracetamol
51	>16–18	M					03h05	5.7 ng/mL Fluoxetine, 2.2 ng/mL Norfluoxetine, 21.0 ng/mL Aripiprazol		Fluoxetine, Norfluoxetine, Aripiprazol, Methylphenidate, Ritalinic acid
52	>18	F			■		14h00	Gliclazide, Metformine		Negative
53	>16–18	F					48h00	Negative		Negative
54	>18	F					08h40	Negative		Negative
55	>18	F			■	■	18h15	17.7 ng/mL Trazodone, 10.7 ng/mL mCpp, 31.3 ng/mL Duloxetine		Trazodone, mCpp, Duloxetine
56	>18	F				■	18h10	Negative		Negative
57	>14–16	F				■	09h00	Negative		1.21 g/L Ethanol
58	>18	F	■			■	48h00	21.0 ng/mL Cocaine, 139 ng/mL BZE, 138 ng/mL EME, 2.6 ng/mL CE		N/A
59	>18	F					04h30	166 ng/mL BZE, 12 ng/mL Cocaine, 10 ng/mL EME, 0.7 ng/mL THC, 0.6 ng/mL OH-THC, 50.6 ng/mL THC-COOH		BZE, Cocaine, EME, THC, OH- THC, THC-COOH, MDMA, MDA
60	>18	F					24h00	Lidocaine		Ibuprofen, Lidocaine
61	>18	F				■	24h00	39.9 ng/mL Sertraline, 126 ng/mL Trazodone, 82.0 ng/mL Diazepam, 21.0 ng/mL Nordiazepam, 5.0 ng/mL Temazepam, 11.0 ng/mL Lorazepam		Sertraline, Trazodone, mCpp, Diazepam, Nordiazepam, Temazepam, Oxazepam, Lorazepam, 7-Aminoclonazepam, Pipamperone, Risperidone, Loperamide, DML
62	>16–18	F	■			■	17h00	Metronidazole, Azithromycine		Metronidazole, Azithromycine
63	>18	F			■		13h00	0.62 g/L Ethanol, 1.4 ng/mL Morphine, 10.2 ng/mL Codeine, 190 ng/mL Desalkylflurazepam, 39.0 ng/mL Diazepam, 20.0 ng/mL Nordiazepam, 3.8 ng/mL BZE, 2.5 ng/mL EME	2.42 g/L	2.18 g/L Ethanol, Morphine, Codeine, Hydrocodone, Flurazepam, Desalkylflurazepam, Diazepam, Nordiazepam, Temazepam, Alprazolam, Zolpidem, Cocaine, BZE, EME, Bisoprolol, Quetiapine, Norquetiapine, Haloperidol, THC-COOH
64	>18	X			■		02h35	Piroxicam		Negative
65	>18	F				■	01h15	0.28 g/L Ethanol	0.47 g/L Ethanol	0.91 g/L Ethanol
66	>16–18	F	■				25h00	Negative		Trazodone, Aripiprazol, OH-risperidone
67	>18	F			■	■	32h30	1.0 ng/mL Morphine, 1.8 ng/mL Codeine		Morphine, Codeine, Norcodeine
68	>18	F			■		04h15	147 ng/mL Trazodone, 1.1 ng/mL mCpp, 67 ng/mL Venlafaxine, 69 ng/mL DMV, 119 ng/mL Bromazepam, 12.1 Alprazolam, Pregabaline, Bisoprolol, Methformine, Gliclazide		Trazodone, mCpp, Venlafaxine, DMV, Bromazepam, Alprazolam, Pregabaline, Bisoprolol, Metoprolol, Methformine, Gliclazide, Amoxicilin, Loperamide, DML
69	>18	F			■	■	44h00	Negative		THC-COOH
70	>16–18	X			■	■	07h05	>300 ng/mL Ritalinic acid		0.75 g/L Ethanol, Methylphenidate, Ritalinic acid
71	>18	F				■	13h00	Negative		Negative
72	>14–16	F					05h00	Negative		Negative
73	>18	F			■	■	17h55	3.8 ng/mL BZE		BZE, Cocaine, EME, Levamisole, Bupropion
74	>14–16	F				■	23h30	Negative		Negative
75	>18	F			■	■	60h00	8.8 ng/mL Bupropion, 21.6 ng/mL Clonazepam, 15.4 ng/mL, 7-Aminoclonazepam		Bupropion OH-risperidone7-Aminoclonazepam Indapamide
76	>18	F		■	■	■	08h00	62 ng/mL Citalopram, 26 ng/mL DMC, 61 ng/mL Nortryptilline, 130 ng/mL Diazepam, 257 ng/mL Nordiazepam, 13 ng/mL Temazepam, 17 ng/mL Oxazepam, >300 ng/mL BZE, 48 ng/mL EME		Citalopram, DMC, Nortryptilline, Diazepam, Nordiazepam, Temazepam, Oxazepam, Lormetazepam, Lorazepam, BZE, EME, Cocaine, CE, THC-COOH
77	>18	F	■			■	12h40	Negative		Negative
78	>18	F				■	05h00	0.61 g/L Ethanol	1.36 g/L Ethanol	1.11 g/L Ethanol
79	>18	F			■	■	10h27	1.43 g/L Ethanol, 9.2 ng/mL Venlafaxine, 65 ng/mL DMV, 617 ng/mL Trazodone		1.19 g/L Ethanol, Venlafaxine, DMV, Trazodone, mCpp

Retrograde exploitation for ethanol is based on an elimination rate of 0.15 g/L/h and for a maximum of 12 h. BZE: benzoylecgonine; CE: cocaethylene; DMC: desmethylcitalopram; DML: desmethylloperamide; DMV: desmethylvenlafaxine; EME: ethylmethylecgonine; m-CPP: meta-chlorophenylpiperazine; MDA: methylenemethxyamphetamine; MDMA: methylenemethoxy-methamphetamine; NA: not available; N°: number; OH-: hydroxy-; STA: systematic toxicological analysis; THC: Δ^9^-tetrahydrocannabinol; THC-COOH: 11-nor-9-Δ^9^-tetrahydrocannabinol.

**Table 2 pharmaceuticals-14-00432-t002:** Blood concentrations.

**ALCOHOL**					
	**n**	**low (g/L)**	**high (g/L)**	**median (g/L)**	**mean (g/L)**
**Ethanol**	13	0.28	1.81	0.80	0.95
**DRUGS**					
	**n**	**low (ng/mL)**	**high (ng/mL)**	**median (ng/mL)**	**mean (ng/mL)**
**Cocaine**	8				
Cocaine	2	12.0	21.0	16.5	16.5
BZE	8	3.8	698.0	147.3	175.9
EME	5	2.5	138.0	10.0	38.8
CE	3	2.6	3.9	3.4	3.3
**Cannabis**	5				
THC	1	0.7	0.7	/	/
OH-THC	1	0.6	0.6	/	/
THC-COOH	5	4.7	50.6	5.8	15.0
**Amphetamines**	3				
MDMA	2	83.0	166.0	124.5	124.5
MDA	2	8.0	17.0	12.5	12.5
Amphetamine	1	2807	2807	/	/
**MEDICATION**					
	**n**	**low (ng/mL)**	**high (ng/mL)**	**median (ng/mL)**	**mean (ng/mL)**
**Antidepressants**					
Citalopram (DMC)	2	19.6 (5.4)	62 (26.0)	40.8 (15.7)	40.8 (15.7)
Bupropion	1	8.8	8.8	/	/
Duloxetine	1	31.3	31.3	/	/
Dosulepine	1	9.0	9.0	/	/
Fluoxetine (NF)	2	5.7 (2.2)	88.2 (263.9)	46.9 (133.1)	46.9 (133.1)
Mirtazapine	1	7.9	7.9	/	/
Nortryptyline	1	61.0	61.0	/	/
Sertraline	2	38.4	39.9	39.2	39.2
Trazodone (mCpp)	5	9.2 (60.0)	617.0 (10.7)	126.4 (5.9)	186.5 (5.9)
Venlafaxine (DMV)	2	9.2 (65.0)	67.0 (69.0)	38.1 (67.0)	38.1 (67.0)
**Benzodiazepines**					
Alprazolam	2	6.1	12.1	9.1	9.1
Bromazepam	2	119	235	177	177
Clonazepam (7-aminoC)	1	21.6 (15.4)	21.6 (15.4)	/	/
Desalkylflurazepam	1	190.0	190.0	/	/
Diazepam	4	37.8	130.0	60.5	72.2
Nordiazepam	4	20.0	257.0	79.6	109.1
Oxazepam	2	3.0	17.0	10.0	10.0
Temazepam	2	5.0	13.0	9.0	9.0
Lorazepam	1	11.0	11.0	/	/
Zolpidem	1	0.8	0.8	/	/
**Neuroleptics**					
Aripiprazol	3	21.0	86.5	34.3	47.3
Amisulpride	1	31.0	31.0	/	/
Quetiapine	1	0.9	0.9	/	/
OH-Risperidone	1	4.7	4.7	/	/
Sulpride	1	4.8	4.8	/	/
**Opioids**					
Morphine	2	1	1.4	1.2	1.2
Codeine	2	1.8	10.2	6.0	6.0
**Others**					
Methylfenidate	1	21.5	21.5	/	/
Rilatinic acid	2	>300	>300	>300	>300

BZE: benzoylecgonine; CE: cocaethylene; DMC: desmethylcitalopram; DML: desmethylloperamide; DMV: desmethylvenlafaxine; EME: ethylmethylecgonine; m-CPP: meta-chlorophenylpiperazine; MDA: methylenemethxyamphetamine; MDMA: methylenemethoxymethamphetamine; NF: norfluoxetine; OH-: hydroxy-; THC: Δ^9^-tetrahydrocannabinol; THC-COOH: 11-nor-9-Δ^9^-tetrahydrocannabinol; 7-AminoC: 7-aminoclonazepam.

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
