# Peer review of "The Interest of a Systematic Toxicological Analysis Combined with Forensic Advice to Improve the Judicial Investigation and Final Judgment in Drug Facilitated Sexual Assault Cases"

_pharmaceuticals, 2021, doi:10.3390/ph14050432_

Round 1
Reviewer 1 Report
In the present submission, the authors sought to obtain more knowledge regarding DFSA, create an efficient DFSA analysis strategy and aanwareness to the judicial authorities concerning the role of recreational, over-the-counter (OTC’s) and prescription drugs, and ethanol can play in DFSA cases. Finally, this paper aims to evaluate the potential impact of Systematic Toxicological Analysis (STA) on the final judicial outcome. This manuscript is well written in all sections, clear and informative.
Some suggestions for improvements:
In the Introduction section the Authors should explain the meaning of STA and additionally explain that this method (STA) can be used with other instruments and not only with high-resolution mass spectrometry. There are some STA applications that use for example (see. The use of GC/MS as the case of acute alimemazine toxicity and chronic administration of alimemazine in pediatric case (l Gomila I. et all. Forensic Science International Volume 2016;266: e18-e22 ) or for the identification and quantification of anabolic steroids in pharmaceutical preparations from the black market (Pellegrini M.et all Ann Toxicol Anal. 2012; 24(2): 67-72).
Please mention and provide some other examples for STA
The Tables and the figures should not appear as images, but Tables as a word file and figures as JPEG file. Please contact the Editor and journal website.
Author Response
Reviewer 1: Some suggestions for improvements:
In the Introduction section the Authors should explain the meaning of STA and additionally explain that this method (STA) can be used with other instruments and not only with high-resolution mass spectrometry. There are some STA applications that use for example (see. The use of GC/MS as the case of acute alimemazine toxicity and chronic administration of alimemazine in pediatric case (l Gomila I. et all. Forensic Science International Volume 2016;266: e18-e22 ) or for the identification and quantification of anabolic steroids in pharmaceutical preparations from the black market (Pellegrini M.et all Ann Toxicol Anal. 2012; 24(2): 67-72).
Please mention and provide some other examples for STA
Response 1: We have added a short paragraph in the introduction concerning STA and focussed on the recommendations of the UNODC concerning STA for DFSA analysis. The above mentioned publications are certainly of interest, but we wanted to focus on the DFSA aspects. We also adapted the materials and method section, to explain that the described methodology is the one used in this study. Of course HRMS is not obliged, but can be useful to ensure an adequate screening on a low volume of sample.
The Tables and the figures should not appear as images, but Tables as a word file and figures as JPEG file. Please contact the Editor and journal website.
Response 2: We will contact the Editor.
Reviewer 2 Report
In their review the authors describe the systematic toxicological analysis combined with forensic advice to improve judicial investigation and final judgment in Drug Facilitated Sexual Assault. Cases illustrating the knowledge about this issue and the main compounds used in order to develop efficient strategies for DFSA analysis are described. The manuscript is interesting and makes a complete overview of the problem. In my opinion it is suitable for publication in the present form.
Author Response
Reviewer 2 Comments and Suggestions for Authors
In their review the authors describe the systematic toxicological analysis combined with forensic advice to improve judicial investigation and final judgment in Drug Facilitated Sexual Assault. Cases illustrating the knowledge about this issue and the main compounds used in order to develop efficient strategies for DFSA analysis are described. The manuscript is interesting and makes a complete overview of the problem. In my opinion it is suitable for publication in the present form.
Response 3: Thanks for these nice remarks.
Reviewer 3 Report
The work is interesting, despite being a small sample of cases. The problem of DSFA seems to be growing and it is really worrying.
In these cases, the usual method should be to analyze samples in a systematic way in order to know as best as possible the state of consciousness of the victim at the time of the attack and the possible influence of substance use in this situation.
It seems clear that the so-called classic drugs, together with alcohol, continue to predominate in consumption and are the majority in cases of DSFA. Studies of this type are useful to know the real state of the question at the present time.
However, there are some issues that would be appropriate to correct or clarify:
- A better description of the substances and the methods used is useful. It is described a screening with LC-HRMS, ethanol analysis with a headspace GC-FID and determination of GHB with GC-MS after alkylation/acetylation with TFAA/HFB-OH. And the authors refer that drugs were quantified via high-pressure liquid chromatographic-tandem mass spectrometric techniques (UPLC-MS/MS) in multi-reaction-monitoring mode (MRM). However, it is necessary to provide more information about determination on other substances (drugs and drugs such as Trazodone, Zolpidem, Amphetamines and metabolites, benzodiazepines, ...). This information is missing, and it only refers to several bibliographic references from the authors themselves (at least 8 references), some from time ago, without further explanation. It would be useful, at least in relation to Benzodiazepines (a large group of drugs with an important relationship usually with DFSA), make any mention of the analysis method
- On the other hand, a question in these situations in Toxicology laboratories is the amount of sample necessary for the analyzes. If several methods are required for the quantification of substances, after a previous screening, a large quantity of sample (blood and urine) would be necessary, and not always it is available in these cases. Have all the substances that were positive in the previous screening been confirmed? If there have been analyzes that have not been quantified, it would be necessary to explain it, because it is important when making a global interpretation of the results.
- The gender abbreviation in the table is confusing. It would be more suitable F (Female), M (male)
- Case 13: the time elapsed since the assault is 68 h (in the text it is said that all the cases are between 1h44 min and 60 h), and even so there are significant concentrations of benzodiazepines that, under normal conditions, should not remain in blood after this period of time. If they are due to medical treatment after the assault, it should be specified because it interferes with the interpretation of toxicological results at the time of the assault.
- Case 58: It seems strange that, after 48 hours, Cocaine (and not a metabolite) is found in the blood. Is this really the case?
Author Response
Reviewer 3 However, there are some issues that would be appropriate to correct or clarify:
- A better description of the substances and the methods used is useful. It is described a screening with LC-HRMS, ethanol analysis with a headspace GC-FID and determination of GHB with GC-MS after alkylation/acetylation with TFAA/HFB-OH. And the authors refer that drugs were quantified via high-pressure liquid chromatographic-tandem mass spectrometric techniques (UPLC-MS/MS) in multi-reaction-monitoring mode (MRM). However, it is necessary to provide more information about determination on other substances (drugs and drugs such as Trazodone, Zolpidem, Amphetamines and metabolites, benzodiazepines, ...). This information is missing, and it only refers to several bibliographic references from the authors themselves (at least 8 references), some from time ago, without further explanation. It would be useful, at least in relation to Benzodiazepines (a large group of drugs with an important relationship usually with DFSA), make any mention of the analysis method
Response 4: It was our aim to describe the applied methods for the study as concise as possible ensuring that the reader could get an optimal information to reproduce the methods if necessary, but without an overload of information concerning the applied materials and methods. Therefore, we referred to the peer-reviewed publication of the applied methods. The method that has not been published yet was described in detail in the method section. However, we have added some details and clarifications in the method section to give the reader more information concerning the type of compounds that we are looking for (see changes in manuscript section Methods and Materials/STA.
- On the other hand, a question in these situations in Toxicology laboratories is the amount of sample necessary for the analyzes. If several methods are required for the quantification of substances, after a previous screening, a large quantity of sample (blood and urine) would be necessary, and not always it is available in these cases. Have all the substances that were positive in the previous screening been confirmed? If there have been analyzes that have not been quantified, it would be necessary to explain it, because it is important when making a global interpretation of the results.
Response 5: The necessary volume is added in the manuscript. We have developed our STA in a way that a minimal volume is necessary for screening (100 µL) and confirmation (900 µL). All the compounds are confirmed and quantified. The results of these quantifications are indicated in Table 2 and 3.
- The gender abbreviation in the table is confusing. It would be more suitable F (Female), M (male)
Response 6: This is adapted as requested.
- Case 13: the time elapsed since the assault is 68 h (in the text it is said that all the cases are between 1h44 min and 60 h), and even so there are significant concentrations of benzodiazepines that, under normal conditions, should not remain in blood after this period of time. If they are due to medical treatment after the assault, it should be specified because it interferes with the interpretation of toxicological results at the time of the assault.
Response 7: The time elapse in the text was an error and is corrected to 68 h. We have discussed the issue of detecting compounds in cases with a long time elapse already in our discussion section. Thanks for pointing case 13 out: we have now added it to the examples in the text: ‘Moreover, long delays can complicate interpretation due to possible intake or administration of substances after the alleged facts, but before sampling, by the victim or administration during first aid medical treatment (e.g., case 13, 23,26,43,52,58 Table 2 and Figure 3).’
- Case 58: It seems strange that, after 48 hours, Cocaine (and not a metabolite) is found in the blood. Is this really the case?
Response 8: The Table (2) also indicated 139 ng/mL BZE (= benzoylecgonine), 138 ng/mL EME (=methylecgonine) and 2,6 CE (= cocaethyleen). The case is also added to the examples of long delays that can complicate interpretation. (see remark above).
Round 2
Reviewer 3 Report
The authors have modified the issues indicated in the first review so that now the article can be accepted for publication